# Sleep in the Natural Environment: A Pilot Study

**DOI:** 10.3390/s20051378

**Published:** 2020-03-03

**Authors:** Fayzan F. Chaudhry, Matteo Danieletto, Eddye Golden, Jerome Scelza, Greg Botwin, Mark Shervey, Jessica K. De Freitas, Ishan Paranjpe, Girish N. Nadkarni, Riccardo Miotto, Patricia Glowe, Greg Stock, Bethany Percha, Noah Zimmerman, Joel T. Dudley, Benjamin S. Glicksberg

**Affiliations:** 1Hasso Plattner Institute for Digital Health at Mount Sinai, Icahn School of Medicine at Mount Sinai, New York, NY 10032, USA; fayzan.chaudhry@mssm.edu (F.F.C.); matteo.danieletto@mssm.edu (M.D.); eddye.golden@mssm.edu (E.G.); jessica.defreitas@icahn.mssm.edu (J.K.D.F.); ishan.paranjpe@icahn.mssm.edu (I.P.); girish.nadkarni@mountsinai.org (G.N.N.); riccardo.miotto@mssm.edu (R.M.); Patricia.Glowe@mssm.edu (P.G.); 2Department of Genetics and Genomic Sciences, Icahn School of Medicine at Mount Sinai, New York, NY 10032, USA; scelzajr@gmail.com (J.S.); gbotwin@gmail.com (G.B.); markshervey@gmail.com (M.S.); bethany.percha@mssm.edu (B.P.); Noah.zimmerman@mssm.edu (N.Z.); 3Institute for Next Generation Healthcare, Icahn School of Medicine at Mount Sinai, New York, NY 10032, USA; info@gregorystock.net; 4The Charles Bronfman Institute for Personalized Medicine, Icahn School of Medicine at Mount Sinai, New York, NY 10032, USA; 5Department of Medicine, Icahn School of Medicine at Mount Sinai, New York 10032, USA

**Keywords:** wearables, biosensors, sleep, Fitbit, Oura, Hexoskin, Withings, cognition

## Abstract

Sleep quality has been directly linked to cognitive function, quality of life, and a variety of serious diseases across many clinical domains. Standard methods for assessing sleep involve overnight studies in hospital settings, which are uncomfortable, expensive, not representative of real sleep, and difficult to conduct on a large scale. Recently, numerous commercial digital devices have been developed that record physiological data, such as movement, heart rate, and respiratory rate, which can act as a proxy for sleep quality in lieu of standard electroencephalogram recording equipment. The sleep-related output metrics from these devices include sleep staging and total sleep duration and are derived via proprietary algorithms that utilize a variety of these physiological recordings. Each device company makes different claims of accuracy and measures different features of sleep quality, and it is still unknown how well these devices correlate with one another and perform in a research setting. In this pilot study of 21 participants, we investigated whether sleep metric outputs from self-reported sleep metrics (SRSMs) and four sensors, specifically Fitbit Surge (a smart watch), Withings Aura (a sensor pad that is placed under a mattress), Hexoskin (a smart shirt), and Oura Ring (a smart ring), were related to known cognitive and psychological metrics, including the n-back test and Pittsburgh Sleep Quality Index (PSQI). We analyzed correlation between multiple device-related sleep metrics. Furthermore, we investigated relationships between these sleep metrics and cognitive scores across different timepoints and SRSM through univariate linear regressions. We found that correlations for sleep metrics between the devices across the sleep cycle were almost uniformly low, but still significant (*p* < 0.05). For cognitive scores, we found the Withings latency was statistically significant for afternoon and evening timepoints at *p* = 0.016 and *p* = 0.013. We did not find any significant associations between SRSMs and PSQI or cognitive scores. Additionally, Oura Ring’s total sleep duration and efficiency in relation to the PSQI measure was statistically significant at *p* = 0.004 and *p* = 0.033, respectively. These findings can hopefully be used to guide future sensor-based sleep research.

## 1. Introduction

Between 50 and 70 million Americans currently suffer from poor sleep [1]. A 2014 study from the Centers for Disease Control and Prevention found that over one third of Americans (34.8%) regularly sleep less than the recommended 7 hours per night [2]. Although the body has remarkable compensatory mechanisms for acute sleep deprivation, chronic poor sleep quality and suboptimal sleep duration are linked to many adverse health outcomes, including increased risk of diabetes [3], metabolic abnormalities [4], cardiovascular disease [5], hypertension [6], obesity [7], and anxiety and depression [8]. Chronic sleep deprivation also poses economical burdens to society, contributing to premature mortality, loss of working time, and suboptimal education outcomes that cost the US $280.6-411 billion annually [9]. However, the underlying mechanisms mediating the adverse effects of poor sleep remain unknown. Diverse factors and complex interactions govern the relationship between health and sleep, and there is likely substantial inter-individual variability. Pronounced gender [10], race [11], and ethnicity differences in sleep-related behaviors are well-established [2].

It is clear that broad, population-level studies of sleep are necessary to understand how lifestyle and environmental factors contribute to poor sleep and to link sleep abnormalities to their attendant negative health effects [12]. It is particularly important to capture individuals’ sleep patterns in natural sleep settings (i.e., at home). However, traditional approaches to studying sleep do not permit these types of studies. Polysomnography (PSG), where brain waves, oxygen levels, and eye and leg movements are recorded, is the current “gold standard” approach to studying sleep. A PSG study typically requires the participant to sleep in a hospital or clinic setting with uncomfortable sensors placed on the scalp, face, and legs. These studies, which remove the participant from his/her natural sleep environment, are not well suited to longitudinal assessments of sleep. They also create issues such as the first night effect, which limit the translatability of laboratory sleep studies to real-life environments [13]. The recent development of clinical grade, at-home PSG tools has enabled quantification of the laboratory environment’s effect on sleep [14]. Such studies have generally confirmed that participants sleep better at home than they do in a lab, although these findings are not universal [15]. 

Even with the availability of the at-home PSG, however, it is unlikely that the use of expensive, cumbersome, single-purpose equipment will promote the kinds of large-scale population studies that can quantify the diverse factors affecting sleep and its relationship to health outcomes. More user-friendly, lightweight, and unobtrusive sleep sensors are needed; ideally these would be embedded in devices that study participants already own. Recently, several companies have developed sub-clinical grade “wearable” technologies for the consumer market that passively collect high frequency data on physiological, environmental, activity, and sleep variables [16]. The Food and Drug Administration classifies these as general wellness products and they are not approved for clinical sleep studies. Due to their passivity, low risk, and growing ubiquity amongst consumers, it is clear that these devices present an intriguing new avenue for large-scale sleep data collection [17]. Combined with mobile application (app) software to monitor cognitive outcomes such as reaction time, executive function, and working memory, these devices could feasibly be used for large-scale, fully remote sleep studies.

This study aimed to determine the feasibility of monitoring sleep in a participant’s natural environment with surveys completed electronically. Specifically, we performed a week-long pilot comparative study of four commercially available wearable technologies that have sleep monitoring capabilities. For the entire week, 21 participants were instrumented with all four devices, specifically Fitbit Surge (a smart watch), Withings Aura (a sensor pad that is placed under a mattress), Hexoskin (a smart shirt), and Oura Ring (a smart ring). To assess the feasibility of a fully remote study relating sleep features to cognition, we also assessed participants’ daily cognitive function via a series of assessments on a custom-built mobile app. None of the four devices we compared in this study had been previously compared head-to-head for sleep and cognition research. Our results highlight some of the key difficulties involved in designing and executing large-scale sleep studies with consumer-grade wearable devices.

The rest of the paper is organized as follows. In Section 2, we describe the literature of related work including state-of-the-art research. In Section 3, we detail the materials and methods employed in this work, including the participant recruitment process, all metrics collected (e.g., device output), and the statistical tests performed. We detail the results from all assessments in Section 4. We discuss the implications of our work as well as limitations in Section 5 and finally conclude the paper in Section 6.

## 2. State-of-the-Art

This study built off of previous work that utilized comparisons of various devices and polysomnography [18,19]. For instance, de Zambotti et al. [19] directly compared the Oura ring with PSG. Correlation matrices from their study show poor agreement across different sleep stages, showing that tracking sleep stages was a problem for the Oura. However, this study concluded that the Oura’s tracking of total sleep duration (TSD), sleep onset latency, and wake after sleep onset were not statistically different than that of PSG for these metrics. The Oura was found to track TSD in relative accordance with PSG in this regard. This suggests that many devices have trouble tracking TSD or participants had trouble wearing devices correctly outside of a monitored sleep lab. 

The biggest question for these devices is, how well do they actually reflect sleep? The current consensus is mixed. For instance, de Zambotti et al. [20] found good overall agreement between PSG and Jawbone UP device, but there were over- and underestimations for certain sleep parameters such as sleep onset latency. Another study compared PSG to the Oura ring and found no differences in sleep onset latency, total sleep time, and wake after sleep onset, but the authors did find differences in sleep stage characterization between the two recording methods [19]. Meltzer et al. [21] concluded that the Fitbit Ultra did not produce clinically comparable results to PSG for certain sleep metrics. Montgomery-Downs et al. [22] found that Fitbit and actigraph monitoring consistently misidentified sleep and wake states compared to PSG, and they highlighted the challenge of using such devices for sleep research in different age groups. While such wearables offer huge promise for sleep research, there are a wide variety of additional challenges regarding their utility, including accuracy of sleep automation functions, detection range, and tracking reliability, among others [23]. Furthermore, comprehensive research including randomized control trials as well as interdisciplinary input from physicians and computer, behavioral, and data scientists will be required before these wearables can be ready for full clinical integration [24].

As there are many existing commercial devices, it is not only important to determine how accurate they are in capturing certain physiological parameters, but also the extent to which they are calibrated compared to one another. In this way, findings from studies that use different devices but measure similar outcomes can be compared in context. Murakami et al. [25] evaluated 12 devices for their ability to capture total energy expenditure against the gold standard and found that while most devices had strong correlation (greater than 0.8) compared to the gold standard, they did vary in their accuracy, with some significantly under- or overestimating energy expenditure. The authors suggested that most wearable devices do not produce a valid quantification of energy expenditure. Xie et al. [26] compared six devices and two smartphone apps regarding their ability to measure major health indicators (e.g., heart rate or number of steps) under various activity states (e.g., resting, running, and sleeping). They found that the devices had high measurement accuracy for all health indicators except energy consumption, but there was variation between devices, with certain ones performing better than others for specific indicators in different activity states. In terms of sleep, they found the overall accuracy for devices to be high in comparison to output from the Apple Watch 2, which was used as the gold standard. Lee et al. [27] performed a highly relevant study in which they examined the comparability of five devices total and a research-grade accelerometer to self-reported sleep regarding their ability to capture key sleep parameters such as total sleep time and time spent in bed, for one to three nights of sleep.

## 3. Materials and Methods

### 3.1. Research Setting

Participants were enrolled individually at the Harris Center for Precision Wellness (HC) and Institute for Next Generation Healthcare research offices within the Icahn School of Medicine at Mount Sinai. Monetary compensation in the form of a $100 gift card was provided to study participants upon device return. During the enrollment visit, participants met with an authorized study team member in a private office to complete the consent process, onboarding, and baseline procedures. The remainder of the study activities took place remotely with limited participant-team interaction. The study team maintained remote contact with each research participant throughout his/her participation via phone or email to answer any questions and provide technical support. The study was approved by the Mount Sinai Program for the Protection of Human Subjects (IRB #15-01012).

### 3.2. Recruitment Methods

To ensure a diverse population, the participants were recruited using a variety of methods, including flyers, institutional e-mails, social media, institution-affiliated websites, websites that help match studies with participants, and referrals.

### 3.3. Inclusion and Exclusion Criteria

Participants were eligible for the study if they were over 18 years old, had access to an iPhone, had basic knowledge of installing and using mobile applications and wearable devices, and were willing and able to provide written informed consent and participate in study procedures. Participants were ineligible for the study if they were colorblind, part of a vulnerable population, or unwilling to consent and participate in study activities. 

### 3.4. Onboarding Questionnaires

During the initial study visit, participants were prompted to complete four questionnaires (see Appendix A). All questionnaires were completed electronically via SurveyMonkey and the results were subsequently stored in the study team’s encrypted and secured electronic database. 

The Demographics Questionnaire (Appendix A) ascertained basic demographic information.

The 36-Item Short Form Health Survey (SF-36; Appendix A). The SF-36 evaluated eight domains: physical functioning, role limitations due to physical health, role limitations due to emotional problems, energy/fatigue, emotional well-being, social functioning, pain, and general health. The SF-36 takes roughly 5–10 min to complete. 

The Morningness-Eveningness Questionnaire (MEQ; Appendix A) is a 19-question, multiple-choice instrument designed to detect when a person’s circadian rhythm allows for peak alertness. The MEQ takes roughly 5–10 min to complete. 

The Pittsburgh Sleep Quality Instrument (PSQI; Appendix A) is a nine-item, self-rated questionnaire that assesses sleep over the prior month. The PSQI has been shown to be sensitive and specific in distinguishing between good and poor sleepers. The PSQI utilizes higher numbers to indicate poorer sleep. The PSQI takes roughly 5–10 min to complete.

### 3.5. Technology Setup and Testing

After the initial screening visit, participants were asked to set up their devices and begin the week-long study at their leisure (Figure 1). The study team chose technologies based on performance and usability data obtained from HS#: 15-00292, “Pilot Evaluation Study on Emerging Wearable Technologies.” Each participant was assigned four sleep monitoring devices: a Fitbit Surge smart watch (Fitbit; first edition), a Hexoskin smart shirt (Hexoskin; male and female shirts and Classic device), a Withings Aura sleep pad/system (Withings; model number WAS01), and an Oura smart ring (Oura; first edition). Note that the form factors for the four devices were different; this was important to ensure that they could all be used at once and would not interfere with each other.

Setup for each device involved downloading the corresponding manufacturer’s mobile application on the participant’s iPhone and downloading the study team’s custom HC App. Participants agreed to each manufacturer’s software terms and conditions in the same manner as if they were to purchase and install the technologies themselves. In doing so, and as noted in the participant-signed consent document, participants acknowledged that the manufacturers would have access to identifiable information such as their names, email addresses, and locations. The HC App functioned as a portal to allow participants to authorize the sharing of data between the manufacturers’ applications and the study team’s database. During the initial setup period, the study team worked with participants to troubleshoot any issues and ensure proper data transmission to the database.

### 3.6. Sleep Monitoring and Device-Specific Parameters

Over a 7-day consecutive monitoring period of the participant’s choosing, participants used the four different sleep monitoring technologies and completed daily assessments (Figure 1). The monitors measured physiological parameters (e.g., heart rate, heart rate variability, respiratory rate, temperature, and movement), activity parameters (e.g., number of steps per day), and sleep-related parameters, specifically time in each sleep stage, time in bed to fall asleep (latency), TSD, number of wakeups per night (wakeups), and standardized score of sleep quality (efficiency). The Withings and Oura both stage sleep as: (1) awake, (2) light, (3) deep, and (4) rapid eye movement (REM; Figure 1). The Hexoskin stages sleep as (1) awake, (2) non-REM (NREM), and (3) REM. The Fitbit stages sleep as (1) very awake, (2) awake, and (3) asleep.

### 3.7. Daily Questionnaires and n-Back Tests

Using the HC App, participants completed questionnaires and cognitive assessments on each day of the 7-day study. These included the n-back test and self-reported sleep metrics (SRSMs).

#### 3.7.1. n-Back Tests

The n-back test [28] assesses working memory as well as higher cognitive functions/fluid intelligence. Participants were prompted to take the n-back test three times per day (morning, afternoon, and evening). In each test, participants were presented with a sequence of 20 trials, each of which consisted of a picture of one of eight stimuli: eye, bug, tree, car, bell, star, bed, anchor. The participant was asked whether the image was the same as the image n times back from the current image, where n = 1 or 2. The stimuli were chosen so that in the course of 20 trials, 10 would be congruent (the stimulus would match the n-back stimulus) and 10 would be incongruent. The participant had 500 milliseconds to enter a response. If no response was entered, the trial was counted as incorrect and a new trial was presented. The n-back tests took roughly 3 min each, for a total of under 10 min/day.

#### 3.7.2. SRSMs

The participant was asked for an estimate of TSD, latency (i.e., time to fall asleep), and start to end sleep duration (i.e., TSD plus latency, referred to as Start-End). Participants self-reported these metrics electronically through the HC App at wakeup (1–2 min completion time).

### 3.8. n-Back Test Scoring

For each trial (i.e., each morning, afternoon, evening per study day), the participants’ response time and the correctness/incorrectness of responses were recorded. We calculated four different scores for the n-back tests: median reaction time and percent correct, stratified by congruent vs. incongruent items. We treated all reaction times the same and did not segment or weigh based on items that the participants got correct vs. incorrect. Each participant was then given a cognitive score based on a self-created scoring function (Equation (1)) of the reaction time, degree of difficulty of question, and correctness. The metric accounts for variation across multiple elements of the n-back results leading to a greater representation of performance. The formula for the metric is
(1)$$\frac{\sum \left(1-\frac{Reaction\;Time}{Max\;Reaction\;Time}\right)*Answer\;Correct*\frac{Steps\;Back}{2}}{n}$$

### 3.9. Inter-Device Comparisons for Sleep Staging and Metrics

We compared each pair of devices for overall correlation in sleep staging across all nights on a per-epoch basis. While the other three devices were used in this analysis, Fitbit was not included because it does not segment sleep by stages, rather measuring asleep vs. not asleep. Oura and Withings track four stages of sleep while the Hexoskin tracks three (see Section 3.6). Accordingly, the NREM sleep stages for Withings and Oura were combined into a single category (NREM) for this correlation analysis. After this transformation, these three devices had three stages of sleep used for this correlation analysis: (1) awake, (2) NREM, and (3) REM. We utilized Kendall’s rank correlation for this analysis as sleep staging was ordinal. We performed Pearson correlation to compare the between-device correlation for specific device-produced sleep metrics, specifically TSD (all four devices) and REM (Oura, Hexoskin, and Withings), both in terms of total seconds. We also assessed the correlation of SRSMs, specifically TSD, to device-produced TSD (all four devices) across all nights per participant. We used Pearson’s correlation for this analysis as density plots of these data did not reveal any outliers (Figure 2).

### 3.10. Statistical Models Linking Device Data to PSQI and n-Back Scores

We built a series of univariate linear models that regressed each individual sleep feature on either PSQI score or n-back score. The PSQI tracks quality of sleep, with higher values indicating poorer sleep. We performed a series of univariate linear regressions on the one-time reported PSQI against all available device and SRSMs (TSD and latency), taking the mean of each metric across all nights of sleep for each participant as a general representation of sleep quality. These device metrics include: latency, TSD (in hours), wakeups (in number of events), efficiency, and REM (in hours). For these analyses, one participant was not included due to lack of data. Additionally, we used univariate linear regressions to compare n-back score against device and SRSM data. For each analysis, we regressed the n-back score of each timepoint (i.e., morning, afternoon, evening) against the mean of each device metric or SRSM feature by participant. In all of the regression models for the n-back scores, we analyzed only participants with two or more days of reported scores for each timepoint. This left us with 16, 19, and 18 participants out of the original 21 for morning, afternoon, and evening n-back tests, respectively. 

### 3.11. Analysis of Missing Data

We analyzed the degree of missingness of each device-reported or self-reported field as measures of device reliability/quality or participant compliance, respectively. As the study progressed, some sleep features were also updated due to new advances in hardware and software on the device side, which resulted in missing data columns that were not included in the missing data plot.

## 4. Results

### 4.1. Summary of Study Population

Table 1 describes our study population, which consisted of 21 participants (11 female; 10 male). The median age of the cohort was 29 years (range: 23–41). The median PSQI score was 4 (range: 1–12). Sixteen of our participants were classified as normal sleepers, three were poor sleepers, and two were very poor sleepers. Median MEQ score was 52 (range: 35–73). We provide score summaries for all eight SF-36 subcategories at the bottom of Table 1. Additionally, the racial breakdown of the cohort was as follows: 17 Caucasian (white) and 4 Asian participants.

### 4.2. Inter-Device Comparisons for Sleep Stages and Metrics

Table 2 shows the summary statistics for all device-produced metrics and SRSMs. TSD was reported by all devices and by the participants themselves (i.e., as part of SRSM). Figure 2A shows a correlation matrix of TSD. The correlations were generally medium to weak (ρ < 0.7 for all pairwise comparisons), although surprisingly the correlations of the SRSM with device estimates were on par with correlations among the devices themselves. Figure 2B shows a REM sleep (in sec) cycle correlations across the Oura, Hexoskin and Withings (Fitbit did not report an estimate of REM sleep). The correlation between Oura and Withings was highest at ρ = 0.44, while Oura and Hexoskin had the lowest correlation (ρ = 0.22). Figure 2C shows Kendall’s rank correlation across overall sleep stages for Withings, Hexoskin, and Oura (see Section 3.9). All of these assessments were statistically significant at the *p* < 0.05 threshold. We report the *p* values from these analyses in Appendix A.

### 4.3. PSQI, Cognitive Scores, and SRSMs vs. Device Data

Table 3 shows the results of a series of univariate linear models, each of which included either PSQI or cognitive score (morning, afternoon, and evening timepoints) as the dependent variable and the mean of the device metric per participant as the independent variable. The only statistically significant associations for PSQI at scale (i.e., significance threshold at alpha = 0.05) were for Oura’s measurement of TSD and sleep efficiency (*p*< 0.05 for both). In both cases, an increase in TSD or sleep efficiency was associated with a significant decrease in PSQI score; since PSQI increases with poor sleep quality, these associations are in the expected direction (more sleep or more efficient sleep leads to better or lower PSQI). Withings latency was statistically significant for afternoon cognitive scores and evening cognitive scores at *p* = 0.016 and *p* = 0.013, respectively. We did not find any significant associations between SRSMs and cognitive scores or overall PSQI.

### 4.4. Cognitive Scores vs. Participant Summary Data

Table 4 shows the results of univariate linear models that regressed cognitive score on participant summary features. With regards to the morning cognitive score, there was a significant association with the SF-36 sub-category of physical functioning (*p* = 0.014); however, further analysis revealed that this was due to the presence of an outlier with very low physical functioning as well as a low cognitive score, and exclusion of this individual removed the significant association. The SF-36 sub-category of emotional well-being was trending towards significance (*p* = 0.078) with cognitive score that appears robust to the removal of individual data points. None of the other summary features were significantly associated with the morning cognitive score. Several other features were statistically significant (*p* < 0.05) across two or more cognitive score timepoints, and also of note is the consensus of significance of features across afternoon and evening cognitive score timepoints. (Table 4).

### 4.5. Correlation between MEQ Preference and Cognitive Test Response Rates

We illustrate the rate of missingness for sleep-related metrics in Figure 3; in general, a large proportion of relevant data were missing due to noncompliance by users or device malfunctioning. We stratified response rate for morning, afternoon and evening test results, which are grouped by participants’ MEQ segmentation into morning, intermediate, and night in Figure 4. We see that morning-preferred participants had the lowest response rate across all times. Furthermore, we see that afternoon response times were the highest for all MEQ groupings. 

## 5. Discussion

The results of our study reflect some general findings that are likely to impact most research involving wearable devices and mobile apps. First, because of low enrollment, our ability to detect effects was low; an effect would need to be highly pronounced to be detectable in a study population of this size. The effort involved in publicizing the study, enrolling participants, and ensuring they were able to complete the study (no device or app malfunctions, no devices running out of batteries, etc.) was substantial. Simple study designs with perhaps one or two devices that participants already own and are familiar with offer the greatest chance of success on a large scale. Second, there was substantial variability among the devices we tested, making the choice of device for any sleep study a material factor that can impact results. Even if it is impossible to assess which device is “preferred” for a given study design, this variability impacts cross-interpretability of results across different studies and will thwart attempts at meta-analyses. In this work, for TSD, we found that Oura, which has been shown to correlate strongly with PSG in prior work [19], had moderate correlations to Fitbit (0.51), Hexoskin (0.37), and Withings (0.50). Additionally, across the three devices that tracked REM, the maximum correlation was only 0.44 between Oura and the Withings. Finally, missingness and the presence of outliers were important considerations for all statistical analyses with this dataset. Although this was a pilot study, all of these issues are likely to translate to larger wearable device studies as well. 

### 5.1. Study Limitations

There were several limitations to this study. First, we did not include other cognitive assessment tests such as the psychomotor vigilance test. Furthermore, while the n-back test is often used as an assessment of working memory in sleep-related research, the particular composite metric we derived to gauge performance has not been previously validated in this regard. A color-word association task based off the Stroop test was given, but we were not able to analyze results due to poor response rate. Additionally, as a result of using commercial sensors, we were unable to fully blind participants to the output of their sleep devices. While the participants were instructed not to check the nightly device sleep metric outputs when recording their estimated SRSMs, this could have led to biased responses if they did so. The biggest limitation of the current study was the lack of a gold standard for sleep metrics, namely PSG. It should be noted that sleep studies are extremely difficult to conduct with large numbers of participants due to the prohibitive cost of PSG. In the future, however, this field can face huge growth if some amalgamation of cheap, at home devices could reliably track various data and cross confirm results amongst themselves. This would be extremely beneficial in creating a mapping function of individual device metrics to PSG metrics, which in turn could allow these more simplistic sensors to accurately recreate conditions of PSG’s at a low cost and in the comfort of participants’ homes. This mapping function could increase recruitment of participants while decreasing cost for sleep studies. 

### 5.2. Considerations Related to Cognitive Metrics and Self-Reported Sleep Quality Indices

PSQI has been shown to be a poor screening measure of PSG [29]. This may explain why self-reported one-time PSQI sleep quality variation was not well explained by much of the device data. However, the Oura ring’s measurements of efficiency and sleep duration did explain variation in the one-time PSQI with statistical significance. These Oura tracking metrics may merit further investigation. Also, it is important to note that poor tracking metrics and a low number of participants could also be the reason more device data was not able to explain variation in PSQI. In terms of the SRSMs, specifically TSD, we found significant (*p* < 0.05), albeit low (range: 0.31–0.58) correlations between all devices. 

Evidence of using the n-back test as a fluid intelligence metric is contentiously accepted, with some critics citing low correlation between n-back and other fluid intelligence tests [30]. The cognition metric, taken from the N-back results, and results from participant summary data had statistically significant associations. This provides a direction for further studies to investigate with larger samples. Ultimately, higher statistical power is needed to help understand these relationships. A recent study showed that poor sleep or deprivation may cause local deficits, specifically for tasks of an emotional nature [31]. This may suggest implementing a metric for wellbeing in addition to fluid intelligence tasks. Of particular note was Withings latency, which was statistically significant for the afternoon and evening cognitive scores (*p* < 0.05). Due to the low sample size, the importance of this is uncertain, but hopefully with subsequent studies could build on this work by further comparing latency with cognitive scores.

Insight from response rate based on MEQ segmentation into three categories (early, intermediate, and late preference) could help future study designs. Across all MEQ groups, n-back test response rates were highest in the afternoon. This suggests that crucial surveys should be administered around this time if possible. Another finding of note is that late-preferred participants had the highest n-back test response rate on average in the morning and afternoon timepoints. This finding suggests that participants who are not late-preferred may need extra motivating factors to increase their response rates.

## 6. Conclusions

We reported correlations among important sleep metrics for four different sleep tracker devices and correlated the results with self-reported questionnaires and cognitive metrics, specifically the n-back. Difficulty in participant enrollment and engagement led to new ideas about recruitment design and participant engagement design. Exploiting existing technology such as ReasearchKit or HealthKit from Apple can have a twofold benefit for recruiting people remotely (with an e-consent feature built into ResearchKit) and sharing electronic health records (EHR). By further combining this with additional data stores present in the HealthApp, participant eligibility screening can be improved [32]. In consideration of the missing data in the questionnaires and active tasks prescribed, we promote the use of as many passive collection procedures as possible. One such option is a smart mirror [33], which can be more passive than using a smartphone for data (e.g., imaging) collection. Finally, the weak correlation among devices opens new challenges for accurate interpretation and data portability for the end user. How will device-specific findings from various studies be taken in context to one another? The results from the current study can hopefully highlight the need for better standardization for sleep-related metrics across devices in order to make any robust and accurate conclusions.

## Figures and Tables

**Figure 1 sensors-20-01378-f001:**
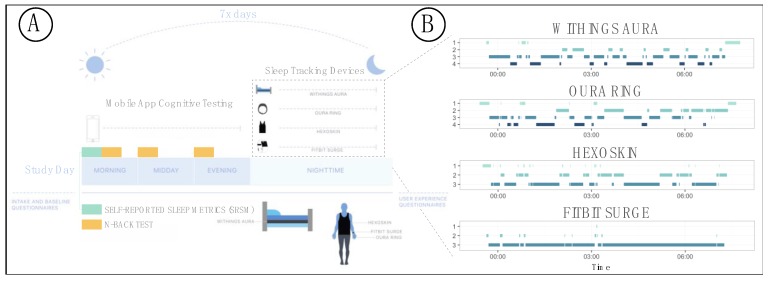
Study structure and data collection for our pilot sleep study. (**A**) Illustration of sleep study monitoring procedure and data collection strategies. (**B**) Example data showing a comparison of sleep staging of a single night for one study participant for all four devices.

**Figure 2 sensors-20-01378-f002:**
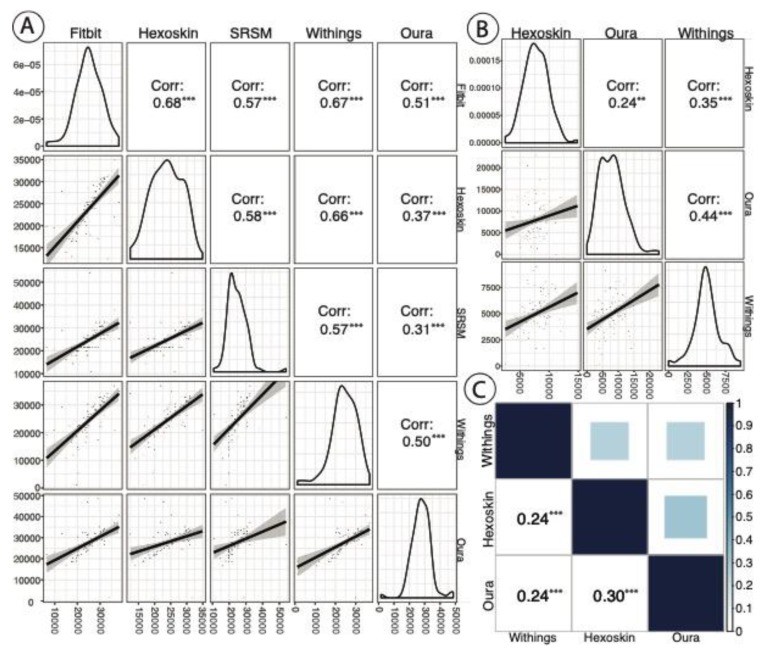
(**A**) A correlation matrix of total sleep duration (TSD) (in seconds) by device and self-reported estimation (i.e., self-reported sleep metrics (SRSMs)) with *p* value significance indication (* *p* < 0.1; ** *p* < 0.05; *** *p* < 0.01). Each point represents data from each night for each participant. The plots in the diagonals of A and B reflect the distribution of sleep metric of interest (TSD and REM, respectively). (**B**) A REM sleep (in sec) correlation across the Oura, Hexoskin, and Withings devices with *p* value significance indication (same as above). The Fitbit was excluded, as it does not track REM vs. NREM sleep. for each individual device. The plots in the bottom left of A and B show the trend line with 95% confidence intervals between devices. (**C**) A correlation matrix of overall sleep stages (awake, NREM, and REM) between Oura, Hexoskin, and Withings devices (Fitbit does not differentiate between NREM and REM) with *p* value significance indication (same as above).

**Figure 3 sensors-20-01378-f003:**
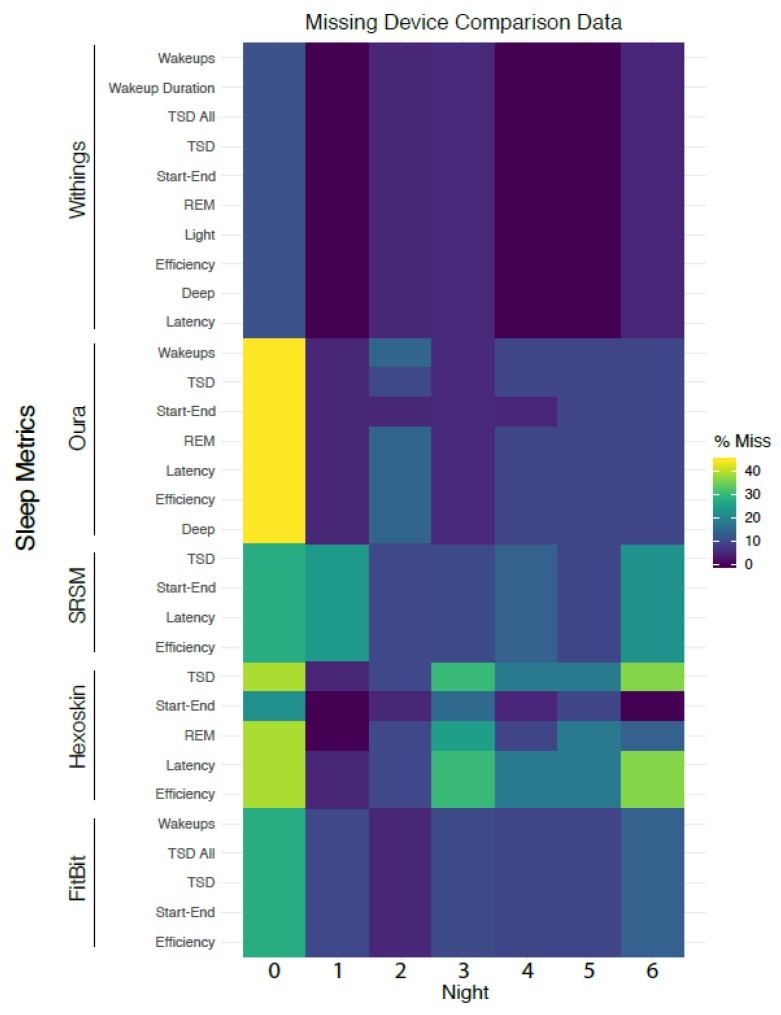
Plot of missing sleep-related data including SRSMs. Due to various device preferences, missing data are asymmetric across devices.

**Figure 4 sensors-20-01378-f004:**
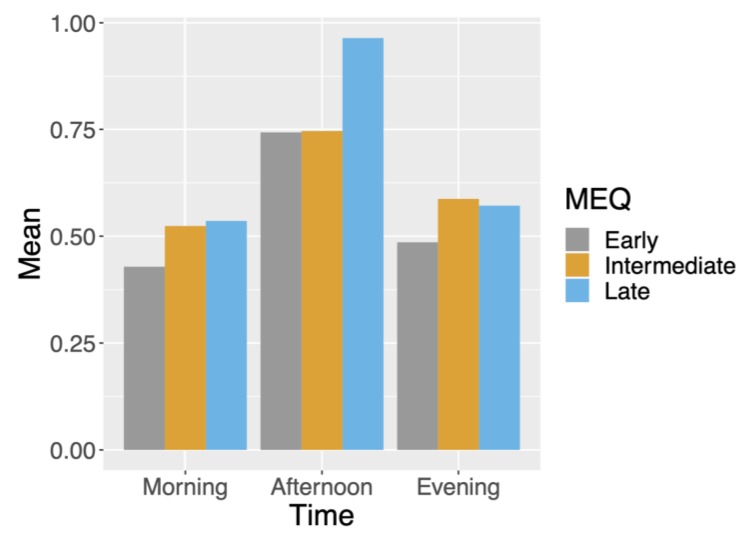
Average missing data for n-back tests by timepoint (morning, afternoon, and evening) and MEQ groupings (early, intermediate, or late).

**Table 1 sensors-20-01378-t001:** Summary of the study population. The participant’s gender (M/F/O), baseline assessment of sleep quality according to the Pittsburgh Sleep Quality Index (PSQI) (with higher values indicative of poorer sleep), age, SF-36 score (a measure of general health along eight axes), and MEQ time (optimal time of day) are included.

ID	Gender	Age	PSQI	MEQ	SF-36 Scores							
					Physical Functioning	Role Limitations (Physical)	Role Limitations (Emotional)	Energy	Emotional Well-Being	Social Functioning	Pain	General Health
1	F	23	1	50	100	100	100.0	50	68	87.5	100.0	55
2	F	26	4	47	90	100	66.7	45	72	100.0	100.0	60
3	F	27	5	52	100	100	100.0	45	56	87.5	90.0	50
4	F	27	2	36	100	100	100.0	65	80	75.0	100.0	55
5	F	27	4	58	100	100	100.0	50	76	87.5	90.0	55
6	F	28	3	52	100	100	100.0	55	76	75.0	100.0	60
7	F	28	3	40	90	100	33.3	50	72	87.5	67.5	55
8	F	29	12	35	100	100	0	15	36	50.0	67.5	55
9	F	31	4	49	95	100	100.0	60	84	100.0	100.0	55
10	F	39	4	49	60	50	100.0	45	44	87.5	77.5	55
11	F	41	5	53	100	100	100.0	95	96	100.0	100.0	60
12	M	25	10	55	100	100	66.7	85	76	100.0	100.0	60
13	M	29	5	52	100	100	100.0	50	88	100.0	100.0	50
14	M	29	4	41	100	100	100.0	50	76	100.0	100.0	60
15	M	31	3	56	95	100	100.0	65	80	75.0	90.0	50
16	M	34	12	73	100	100	66.7	50	52	62.5	100.0	55
17	M	35	6	52	100	100	100.0	75	80	100.0	100.0	50
18	M	37	3	61	90	100	66.7	50	80	87.5	90.0	55
19	M	39	8	72	100	100	100.0	80	88	100.0	100.0	55
20	M	41	6	55	95	100	100.0	50	84	100.0	80.0	55
21	M	41	9	52	95	100	66.7	35	52	87.5	70.0	60
MIN		23	1	35	60	50	0	15	36	50	67.5	50
MEDIAN		29	4	52	100	100	100	50	76	87.5	100	55
MAX		41	12	73	100	100	100	95	96	100	100	60

**Table 2 sensors-20-01378-t002:** Summary metrics of device data and SRSMs. All units are in hours except wakeups which is in occurrences and efficiency (no units). Sleep efficiency is a metric to track percentage of time in bed while asleep. TSD is total sleep duration which is similar to start-end duration and similar features were utilized that included latency and other measures.

Device	Metric	n	Mean	St. Dev	Min	Pctl (25)	Pctl (75)	Max
Fitbit	Efficiency	129	94.70	15.70	31.00	94.00	97.00	193.00
	TSD All	129	7.47	1.47	3.78	6.50	8.43	11.40
	TSD	129	7.58	1.58	1.78	5.98	7.93	10.75
	Start-End	129	7.58	1.73	3.78	6.50	8.48	15.87
	Wakeups	129	1.60	1.20	0.00	1.00	2.00	8.00
Hexoskin	Efficiency	114	92.40	4.40	70.30	91.10	95.30	97.80
	TSD	114	6.72	1.31	3.45	5.78	7.81	9.69
	Start-End	135	7.57	1.42	3.93	6.57	8.58	11.43
	REM	123	2.15	0.57	0.69	1.77	2.53	4.12
	Latency	114	0.29	0.26	0.07	0.12	0.38	1.56
Oura	Efficiency	127	89.70	14.40	24.00	84.00	93.00	164.00
	TSD	128	7.69	1.72	0.42	6.73	8.75	13.48
	Start-End	130	10.67	11.63	4.62	6.97	9.55	117.60
	REM	127	2.17	1.11	0.00	1.29	2.81	6.38
	Deep	127	1.12	0.58	0.00	0.73	1.44	2.58
	Wakeups	127	2.40	1.90	0.00	1.00	4.00	7.00
	Latency	127	0.26	0.25	0.01	0.11	0.30	1.58
Withings	Efficiency	141	84.10	20.50	20.50	74.80	90.10	179.80
	TSD All	141	8.99	2.89	0.53	7.45	10.12	27.03
	TSD	141	6.97	1.75	0.33	5.95	8.15	10.97
	Start-End	141	9.30	4.45	0.42	7.08	9.73	34.55
	REM	141	1.40	0.46	0.00	1.15	1.67	2.63
	Deep	141	1.74	0.58	0.00	1.42	2.15	3.67
	Light	141	3.83	0.98	0.33	3.22	4.45	6.03
	Wakeups	141	2.40	2.60	0.00	0.00	3.00	13.00
	Latency	141	0.32	0.36	0.00	0.08	0.42	2.37
	Wakeup Duration	141	1.38	2.14	0.03	0.53	1.50	17.48
SRSMs	Start-End	122	7.34	1.45	4.50	6.35	8.24	12.33
	TSD	122	6.91	1.56	3.00	6.00	7.78	15.00
	Latency	122	0.24	0.23	0.02	0.08	0.33	2.00

**Table 3 sensors-20-01378-t003:** Results of multiple univariate linear models for PSQI (left) and cognitive scores across all timepoints (right). For the PSQI-related models, the independent variables were the means of device data for each participant, and the dependent variable was PSQI. The higher the value is on the PSQI, the worse the sleep quality; thus, positive correlations suggest relation to poorer sleep quality. For the cognitive score-related models, the independent variables were the means of device data for each participant, and the dependent variable were the cognitive scores. We show the *p* values of each univariate regression for cognitive score by timepoint. Please see Appendix A for more statistics related to these regressions. All units are in hours with the exception of wakeups (number of occurrences) and efficiency (a standardized metric).

		PSQI				Cognitive scores (*p* Value)		
Device	Feature	Coefficient	Std. Error	*p*-Value	R2	Morning	Afternoon	Evening
Fitbit	TSD	−0.273	0.544	0.622	0.014	0.825	0.511	0.610
	Wakeups	1.570	1.005	0.136	0.119	0.329	0.672	0.857
Withings	TSD	−0.125	0.498	0.804	0.004	0.110	0.497	0.409
	Latency	−2.080	2.83	0.472	0.0291	0.869	0.016 **	0.013 **
	Efficiency	0.010	0.060	0.869	0.002	0.315	0.148	0.194
	Wakeups	0.352	0.427	0.421	0.036	0.888	0.361	0.378
	REM	0.260	1.962	0.896	0.001	0.342	0.617	0.557
Oura	TSD	−1.004	0.305	0.004 ***	0.376	0.265	0.197	0.221
	Latency	−7.311	4.445	0.117	0.131	0.366	0.499	0.563
	Efficiency	−0.092	0.040	0.033 **	0.228	0.285	0.332	0.301
	Wakeups	0.168	0.491	0.736	0.006	0.226	0.184	0.289
	REM	−0.526	0.715	0.471	0.029	0.656	0.732	0.713
Hexoskin	TSD	0.187	0.702	0.793	0.004	0.206	0.289	0.235
	Latency	1.249	4.444	0.782	0.004	0.995	0.481	0.718
	Efficiency	−0.226	0.272	0.417	0.037	0.530	0.527	0.798
	REM	0.397	1.833	0.831	0.003	0.128	0.186	0.180
SRSM	TSD	−0.725	0.558	0.210	0.086	0.725	0.361	0.273
	Latency	1.846	4.033	0.653	0.012	0.935	0.210	0.261
Observations		20				16	19	18

Note: * *p* < 0.1; ** *p* < 0.05; *** *p* < 0.01.

**Table 4 sensors-20-01378-t004:** In this collection of univariate linear models, the participants’ summary data are the independent variables, and cognitive score is the dependent variable. We present the *p* values of each univariate regression for cognitive score by timepoint. Please see Appendix A for more statistics related to these regressions. These metrics all represent standardized scores.

		Cognitive Scores (*p* Value)	
Feature	Morning	Afternoon	Evening
PSQI	0.531	0.083 *	0.057 **
MEQ	0.529	0.057	0.120
Emotional Role Limitations	0.665	0.005 ***	0.003 ***
Energy	0.700	0.018 **	0.010 ***
General Health	0.769	0.823	0.961
Physical	0.014 **	0.745	0.597
Social	0.170	0.004 ***	0.002 ***
Well-being	0.078 *	0.005 ***	0.001 ***
Observations	16	19	18

Note: * *p* < 0.1; ** *p* < 0.05; *** *p* < 0.01.

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
