# Peer review of "Sleep in the Natural Environment: A Pilot Study"

_sensors, 2020, doi:10.3390/s20051378_

Round 1

Reviewer 1 Report

In this paper, authors present a pilot study with the goal of investigation of sleep in the natural environment.

Abstract section could be more specific in the sense of the description of the methods and sensors which were used for the analysis. Authors should at least mention what these sensors detect. Also, all the abbreviations should be justified prior the first using.

The introduction section should be aimed on the theoretical background for the study, definition the research hypothesis, which will be consequently justified within the paper. The last part of the introduction section should be dedicated to a brief overview of individual sections.

The consequent section should be dedicated to the state-of-the-art section, where authors should describe the recent methods in this area.

Authors should consider also other evaluation parameters apart from the correlation index which measures a level of linear dependency – agreement. I would recommend employing some difference evaluation techniques like is MSE, RMSE or Euclidean distance.

Regarding the statistical evaluation authors could consider applying the statistical testing for individual tested devices.

Reviewer 2 Report

The study was meant to evaluate a number of commercial digital devices for sleep quality monitoring, by comparing the outputs of the devices to a self-reported PSQI and a score based on N-back test performance.  They found that sleep metrics extracted from those devices did not predict PSQI and N-back metrics.  The correlation between sleep measures among the devices was low in overall. Some associations between outputs of Oura Ring’s and PSQI were found. Although the results could signify a contribution to future sleep studies, the manuscript is not clear about the methodology implemented. Nevertheless, the authors used a metric of cognitive function that has not been previously validated.

Main comments:

Why was N-back test selected? Why not PVT or Stroop test? PVT has been shown to provide good assessment of subjects’ fatigue (e.g. http://www.sleepdisordersflorida.com/pvt1.html) and preparedness-to-perform.  There are indices from PVT that have been previously validated as sensitive to sleep loss (https://www.ncbi.nlm.nih.gov/pubmed/21532951, https://www.ncbi.nlm.nih.gov/pmc/articles/PMC3197786/). Stroop task is also a very sensitive test for assessing subjects’ cognitive function. The derived N-back metric could be sensitive and correct, but it is not previously validated for sleep loss. More clarity on the description of the correlation analysis is required. The different devices provide different categories of sleep, so it is necessary to explain how the measures form the different devices are quantified for correlation analysis. It is not clear what “per participant-night” means.  Also, this sentence is confusing “Additionally, we regressed the N-Back created score to show variance in cognition score across all mean device data by subject and self-reported data”.  As said, in general the description of the correlation analysis requires more clarity. The definition of hit and miss seem to be inadequate. Those terms are commonly reserved to the correct or incorrect response of the subject. Instead, you could use terms like “congruent” or “incongruent” instances.  In fact, you use that terminology later in the manuscript. Did the participants have access to the sleep estimate of the devices? This could have biased their responses to PSQI. Last paragraph of the introduction overlap with methods, results, discussion and conclusion. It is largely unnecessary. Many details (link to websites that will go obsolete, for instance) of the recruiting process are irrelevant. The diversity of the recruited population is what should be shown.  A simple sentence like the following could suffice “The subjects were recruited using a variety of methods, including social media, social media ads, websites, and referrals, to ensure a diverse population.” Although informative, Figure 1 should be redesigned, as it is hard to read and understand. Explain this better “Because we were interested only in concordance between devices, our calculations did not take into account groups of nights corresponding to the same participant.” “We regressed the one-time reported PSQI on all available device metrics” As far as I know, the one-time PSQI was taken before the test, so why did you check for correlation of future sleep with previously-reported sleep quality? Do you mean the daily-taken PSQI? If so, why to use the mean of each metric? What are the metrics in this case? Did you test the statistical significance of the Pearson’s correlation coefficient? Please explain what are those traces in the diagonal of Figure 2. Explain how was computed the significance of associations between features and PSQI and Cognition score. It seems like this is equivalent to Pearson’s correlation analysis (R = Pearson’s correlation coefficient, and you reported R^2)

Other comments:

Need to define PSQI in the abstract. Signification associations or significant associations at scale or at large scale Typo in “strop test”. There are several instances of different formats in the manuscript (line spacing).

Round 2

Reviewer 1 Report

I have rechecked the modified version of the paper. It is apparent that the paper is significantly improved, and I am satisfied with the provided corrections.

Reviewer 2 Report

The authors addressed most of my comments and suggestions, and the quality of the paper is higher in the overall.  A few comments remain unresolved:

  1. From the sentence in lines 285-287, I find confusing the term “mean device”
  2. I couldn´t find in the guidelines the requirement of a “final paragraph that gives a brief breakdown”. Personally, I find such metatext largely unnecessary; although I recognize that the other reviewed suggested doing so.
  3. Given the characteristics of figure 1 (illegible), I suggests making it full-page size.
  4. In figure 2, I suggest using a marker of statistical significance (*, for instance, for p<0.05), instead of including the P value. This would help the clarity of the figure.  In addition, two significant figures should be enough for correlation.
  5. The authors forgot to comment on the observation that the metric of cognitive function obtained from on n-back in this study (equation 1) that has not been previously validated.
